# What Competencies Does a Community Occupational Therapist Need in Neurorehabilitation? Qualitative Perspectives

**DOI:** 10.3390/ijerph19106096

**Published:** 2022-05-17

**Authors:** Daniela Avello-Sáez, Fabiola Helbig-Soto, Nayadet Lucero-González, María del Mar Fernández-Martínez

**Affiliations:** 1Occupational Therapy School, Department of Health Sciences, Faculty of Medicine, Pontificia Universidad Católica de Chile, Santiago 8320000, Chile; daniela.avello@uc.cl (D.A.-S.); nlucero@uc.cl (N.L.-G.); 2Occupational Therapy School, Faculty of Psychology, Universidad de Talca, Talca 3460000, Chile; fabiola.helbig@utalca.cl; 3Department of Sociology, Social Work and Public Health, University of Huelva, 21004 Huelva, Spain

**Keywords:** occupational therapy, neurological rehabilitation, community integration, social inclusion, professional competence

## Abstract

More than three million people in Chile suffer from neurological conditions, and many of these become permanent users of health services with a community approach. In this way, disciplinary competencies in this area are relevant. We seek to characterize the competencies for community occupational therapy intervention in neurorehabilitation. Using a qualitative approach, interviews were conducted with eighteen professionals and were analyzed using content analysis. The main results are associated with the competencies of knowledge associated with theoretical biomedical and community elements. Skills range from health evaluation and intervention on micro- and macrosocial levels. Attitude is also an important skill, stemming from personal and relational spheres. These findings suggest that interventions are essentially on a personal and microsocial level, focusing first on pathology and treatment, and later comprehending the interactions with a patient’s close social environment, such as family, schoolmates, and workmates and their physical environment at home, school, and the workplace. Although the final objective of community intervention is present in the discourse as being able to generate structural changes that favor well-being and social inclusion, concrete competencies are not appreciated on a macrosocial level.

## 1. Introduction

### 1.1. Neurological Disease and Occupation

Approximately one in every six people suffer from a neurological condition in the world, which includes side effects of acquired brain damage, neuropathies, cerebrovascular illnesses, neurological infections, Parkinson’s disease, epilepsy, Alzheimer’s and other dementias, among others. Therefore, this figure represents more than three million people in Chile [1,2].

These conditions impact the occupational performance of people in activities of daily life, the productivity associated with study and work, and in their free time and leisure, significantly reducing their quality of life and that of their families and caregivers [3,4,5].

### 1.2. Competencies in Occupational Therapy

Competencies are defined as a set of technical skills, knowledge, clinical reasoning, and reflection of daily practice for the benefit of the individual and their communities.

The Pan American Health Organization [6] indicates that the identification of health competencies is a high priority, since society is informed about what professionals must know and do. They must have the capacity to use in practice, the knowledge, skills, attitudes, values, and abilities of the profession in the prevention and resolution of health problems [6,7]. 

According to the World Federation of Occupational Therapists, occupational therapy is a client-centered health profession, concerned with promoting health and wellbeing through occupation. The primary goal of this discipline is to enable people to participate in everyday-life activities. To achieve this goal, occupational therapists work with individuals, in an individual and collective manner, by performing interventions to enable their participation in meaningful occupations, enhancing their skills and/or adapting the environment or characteristics of the occupation as needed [8].

There is a series of international publications on occupational therapist profiles and competencies [9,10,11,12,13,14], but there are no specific studies regarding the competencies of the community approach toward neurorehabilitation. However, the World Federation of Occupational Therapy [15] indicates that professional abilities can add value to community interventions, favoring the approach to problems from different perspectives and facilitating the processes from planning to implementation. In line with this evidence, occupational therapists have adhered to practice that is centered on community.

### 1.3. Community Occupational Therapy

The practice of occupational therapy in communities is a methodological model that contributes to guiding collective strategies [16] and has the particularity of having specific knowledge of the occupation as a determinant of health that impacts wellbeing [15]. 

Therefore, this community approach includes characteristics of a community’s territory, networks, and key actors in the design and implementation of clinical care. This approach focuses on the participation of people in their health processes [17] by linking the community’s capacity to systematically and strategically integrate health-related actions with services located in a given territory [18].

In this manner, occupational therapists work to favor people’s participation in collective occupations that promote community health [14,19,20] and social inclusion, committed to the rights of all people. The purpose of these practices is to encourage users to build their future through occupation, as active members of their family, community, and social life, as well as carry out their different roles and responsibilities [21,22]. It is also expected that populations and communities can generate instances of diversity acceptance and promote accessible spaces in physical, social, and cultural contexts [23]. 

Consequently, to carry out meaningful and effective processes, an occupational therapist must develop different competencies that allow him/her to facilitate positive, creative, and culturally relevant environments, such that the communities themselves can address their own problems [14,19,20].

In Latin America, a paradigm shift is underway regarding occupational therapy practice since cultural belonging has been identified as deriving from the knowledge and recognition of a territory’s history. To visualize people holistically, in addition to being a practice centered on the client, a practitioner should not forget that there are structural determinants that can have categorical impacts on health situations and can therefore hinder rehabilitation processes [24].

One of the theoretical references used to support community processes in occupational therapy is Paulo Freire. This reference is used both because of its emancipatory character and its participatory methodologies that facilitate reflective and process awareness about the socio-political conditions that influence the quality of life of users and their community [25,26]. Paulo Freire’s work enriches the professional practice of occupational therapists who seek the active participation of communities in their change processes. As Nunes and Esquerdo [25] mention, ‘Freire’s concepts and proposals can support a critical professional approach intended to engender vulnerable populations and social transformation movements in the dialectical relationship between micro- and macrosocial aspects’.

### 1.4. Experiences of Community Occupational Therapy

In the literature, there are several studies with positive results regarding experiences and research about professional practice in community occupational therapy in neurorehabilitation and its impact on people’s health. 

Among them is a study on non-pharmacological treatments (NPT). These treatments have emerged as a solution to manage certain clinical symptoms that do not necessarily require the consumption of chemicals to improve health in users with mild cognitive impairment and dementia. In addition, NPTs have few side effects and can be used with other treatments. Hence, NPTs have been shown to have a positive impact on cognition, wellbeing, mood, and quality of life in users with neurodegenerative pathologies [27]. 

Vibholm, Christensen, and Pallesen [28] conducted a systematic review on the benefits of nature-based rehabilitation for adults with acquired brain injury, finding evolutionary activity and ecological approaches, among others, as the main theoretical foundations. Social and therapeutic horticulture has also been part of procedural intervention strategies. It is emphasized that this type of intervention must be appropriate for each case. 

Öst Nilsson et al. [29] described the intervention process during nine months of occupational therapy in workers with a history of stroke, theoretically based on a person-centered approach to improve the potential, performance, and level of reentry into the participant’s workforce. At the procedural level, this study considered both the individual and his/her community environment since co-workers intervened and modifications were made to the workplace. From an attitudinal point of view, the professionals who performed the rehabilitation agreed on the objectives and interventions with the patients. After a three-month follow-up, the results indicated that the participant’s performance in the work environment was favorable upon returning to work. 

In addition, another study focused on the relationship between environmental factors and the social participation of children with cerebral palsy (CP). These factors include elements such as physical environments and institutions showing that community and social attitudes play a relevant role in encouraging patient participation. Furthermore, negative attitudes may prevent parents or caregivers from wanting to accompany or encourage the child to move independently or with assistive devices, ultimately reducing their social participation. Generally, environmental modifications and adaptations at home, in the community, and at school can increase the social participation of these children [30]. 

In Australia, Kendall et al. [31] detailed the transition of people with spinal cord injuries from hospital to home in rural areas, where they are referred to local health services. In this situation, patients indicated that the transition was challenging, overwhelming, and complicated because moving from a professional to a not-so-professional place, according to their perceptions, generated feelings of uncertainty. Considering this, the urgency of generating participatory processes, connecting people with local health services before the transition, and providing specialized education on these local devices is recognized. As an attitudinal element, this study emphasizes that the approach should be carried out through a multidisciplinary team, and from a cultural perspective, understanding and facilitating experiences of transition, considering aspects that are specific and unique to the rural environment [31]. 

To bring rehabilitation closer to communities and increase accessibility after hospitalization, following a community health model, the World Health Organization (WHO), developed the community-based rehabilitation (CBR) strategy, which seeks to promote inclusive local development. Another aim of this strategy is to generate a comprehensive rehabilitation process for individuals, their families, social organizations, communities and different government and non-governmental agencies in health, education, labor and social spheres, among others [21]. Public health adheres to a holistic model of family and community health care, moving toward the implementation of territorial and multidisciplinary health teams, where occupational therapists can be found in addition to a community-based rehabilitation network at all levels [32]. 

There is evidence that the CBR strategy has been beneficial for people who have had neurological diseases, especially for early access and a comprehensive approach to their healthcare needs [33,34,35,36]. These conclusions are supported by the fact that CBR professionals are in a continual training process, as mentioned by Guajardo et al. [37], in a study carried out in Chile. This study demonstrated that in rehabilitation, at a theoretical and practical level, professionals are well trained. The latter, along with the strategies and techniques used, allows them to reach their users and obtain positive results in their rehabilitation. However, regarding knowledge linked to the community and territory where they work, these aspects must continue to be strengthened. 

Another study carried out by González-Bernal et al. [38] noted that community occupational therapy plays an important role in both physical and social spheres. This approach highlights promoting the equality of conditions, as well as the promotion of decision making and reduction of exclusion due to disability, with a practical, client-centered model.

Finally, Bianchi and Serrata [39] carried out qualitative research on the professional performance of Latin American community occupational therapists. Their results indicate that practices are articulated at the micro- and macrosocial levels, involving the strengthening of social networks and construction of bonds through occupation. 

In summary, there are several publications on the experiences and foundations that support interventions in community occupational therapy and the field of neurorehabilitation. However, specific competencies that an occupational therapist should possess for a community approach intervention have not been explored in professional practice, which is why the objective of this study is to characterize competencies for community occupational therapy intervention in neurorehabilitation using a qualitative perspective. 

## 2. Materials and Methods 

### 2.1. Study Design

This study has a phenomenological paradigm with a qualitative approach (Figure 1), using a cross-sectional and descriptive scope [40]. It pretends to understand, describe and explain social phenomena: in this case, community intervention and how occupational therapists build the world around them. Importantly, this study seeks to understand what they know, do, or feel and what happens during neurorehabilitation in their own terms [41]. 

Qualitative research emphasizes exploring the nature of a particular phenomenon, investigating its narrative and the meanings that define these practices, while disaggregated in their knowledge, skills, and attitudes. 

### 2.2. Research Context

The research context can be understood as a series of properties that are part of a research phenomenon and the particular condition under which action and interaction strategies are taken to direct, manage, carry out, and respond to this phenomenon [42,43]. 

This study is located in Chile, Latin America, where occupational therapy is a relatively new discipline, officially created in 1963. Currently, there are 3244 occupational therapists in the country. As a reference, there are 114,240 in the United States, 5000 in Spain, and 16,009 in Australia [44].

Chile has a mixed health care system; it is both public and private. In the former, care is provided on three different levels: primary, secondary, and tertiary healthcare. This model includes neurorehabilitation.

Multidisciplinary teams address neurorehabilitation. In primary care, it is provided in rehabilitation wards that offer services to people with chronic or transitory pathologies. In secondary care, there are specialized community centers, both physical and psychosocial, that assist people who require more significant support in their rehabilitation. Finally, some hospitals carry out an early approach in tertiary care and then refer patients to primary or secondary care, depending on the user’s needs to maintain continuity of services [45].

Many of the users that attend these services become permanent beneficiaries since neurological rehabilitation is not temporary, but requires continuous care throughout the entirety of a patient’s lifespan [46]. It is transcendental that professionals in this area contribute to necessary interventions based on the needs of their clients, and these professionals need to have specific core competencies. 

### 2.3. Participants

The study sample was selected through purposive sampling [40]. The inclusion criteria for occupational therapists were: at least three years of experience and consent to cooperate. The data saturation criterion was used to define the qualitative sample, meaning that the number of interviews is determined by the variability of the information obtained, and when the data begin to be reiterated in the discourse of the participants, or the new interviews did not present new ideas, the sample was concluded. This was because the prediction and expectation of the data categories were confirmed repeatedly [41]. 

The sample was made up of eighteen Chilean occupational therapists (Table 1) linked to the area of community intervention in neurorehabilitation of children and adults, most of them working in the physical health area, where they also intervene with a wide range of pathology types, including traumatological, rheumatological, oncological patients, among others. 

Fourteen of them were women between 31 and 55 years of age, and four were men between 26 and 42 years of age. Their work experience ranged from 4 to 22 years, working at the three levels of healthcare in both the public and private systems. In these contexts, occupational therapists are incorporated into a multidisciplinary rehabilitation team. Depending on healthcare level, the patients are classified as in a severe state or in rehabilitation. 

### 2.4. Data Collection

For information gathering, a semi structured, in-depth interview guideline (Table 2) and initial operationalization of the code system (Appendix A) [47,48] were validated by a group of experts in the field. One of the researchers was responsible for conducting the interviews with each participant. The interviews were recorded and carried out at the interviewee’s workplace with an approximate duration of 90 min each, and a field note was prepared for each participant. The interview script could be found in Appendix A.

### 2.5. Data Analysis

The interviews were transcribed in a literal way using the transcription convention of Kvale [49]. All information susceptible to transcription was analyzed through directed content analysis [48] using a tree diagram to order categories and subcategories and through a deductive coding process with a current codebook [50] based on evidence and existing bibliographic information. This was complemented inductively through emerging codes, which were not considered in the initial coding system. Similar codes were grouped together into subcategories, and the subcategories generated a category. 

The categorization complies with the criteria of completeness, where any unit must be able to be in one of the categories, covering all possible data units. In mutual exclusion, each unit is included in a single unit, in which a segment of differentiated text cannot simultaneously belong to more than one category. In the only classificatory principle, each of the categories must be elaborated from a single ordering criterion and classification [51].

Data analysis was performed independently by two researchers, and to ensure data interpretation objectivity, the results were discussed with the research team. The process was assisted by NVivo 12 computer software (QSR International Pty Ltd., Melburne, Australia). The quality criteria in qualitative research indicated by Flick [47] were used, which are rigor and creativity, uniformity and flexibility, and final criteria and strategies.

### 2.6. Ethical Statement

All ethical elements of this research were safeguarded through a process of informed consent, and the names and identities of the participants and the institutions where they work were anonymized. Similarly, the Code of Good Practice of the International PhD School at the University of Almería, Spain, [47] the institution that hosted this project, was followed, which states that given the nature and form of data collection through interviews, this research is exempt from the ethics committee.

## 3. Results

The results of this investigation follow the logic of qualitative data analysis, with each section being one of the categories associated with the competencies identified by occupational therapists in a community neurorehabilitation intervention. 

In general, it can be indicated that these categories are founded on theoretical competencies of health conditions, as well as the community approach; procedural competencies regarding evaluation and intervention on the personal level, in interaction with their physical environment, as well as on a micro- and macrosocial level; and attitude competencies, related to occupational therapists’ personal and relational characteristics. In a generic approach to the participants’ discourse, some highlights are the first ten most cited phrases and words (Figure 2), with a count of 80 repetitions of ‘intervention strategies’, 70 of ‘community approach’, 56 ‘productive roles’, and 50 ‘knowledge’, 49 ‘family’, 46 ‘participatory process’, 45 ‘techniques’, 43 ‘childhood’, and 42 ‘educative area and activities of daily living’. 

Competencies were grouped into three main categories based on bibliographical review [6,7]: knowledge with 39 references, skills with 103, and attitude with 20, for a total of 162 references in the text (Figure 3 and Figure 4). Each reference is a count of the number of selections that have been coded in the different codes. The same selection coded for two different codes is counted as two references 

The importance of determining code frequency in their respective subcategories and categories makes exploratory data observation possible. The content of the discourses presented by the occupational therapists begins to take on similar elements that are grouped into predefined and emerging categories, to reduce the information for analysis, which are analyzed in greater depth in the following sections.

### 3.1. Knowledge

The processes of community intervention in neurorehabilitation are based on strong foundations. These competencies refer to the theoretical knowledge that occupational therapists have, based on the available evidence and bibliographic references. The knowledge includes generic elements of different life course stages, as well as biomedical sciences such as structures and functioning of the central, peripheral, and autonomic nervous system, physiology and pathophysiology, diagnostics, clinical manifestations, and prognosis of neurological pathologies. 

These conceptual elements represent the basis that professionals rely on for the choice and justification of evaluations and interventions. *‘The first thing is knowledge of the main neurological pathologies, such as hemiplegia, cerebrovascular accidents, head injuries, Parkinson’s disease, etc.’**‘To know all the structures and functions of the central, peripheral, and autonomic nervous system.’*

They also provided a theoretical positioning in terms of reference frameworks and conceptual models, which are both clinical and community. There are, for example, motor control theories that aim to inhibit muscle tone patterns and abnormal or pathological reflexes, through sensory stimuli in muscles and joints to facilitate normal movements. This is complemented with the biomechanical frame of reference, based on the principles of kinematics and movement, considering the range of motion, muscle strength, and resistance, incorporating elements of biophysics through the manufacture of orthoses, in which the aim is to increase or maintain mobility. *‘It is important for neurorehabilitation to know about motor control, activity-oriented motor learning, neuronal plasticity, neuromuscular facilitation, the Bobath concept, which is the one that is most used.’**‘You need to know about the biomechanical model, about biomechanics as such, to know the angles of the body, the principles of orthotics for making orthoses.’*

In pediatrics, knowledge of psychomotor development and the respective developmental milestones also becomes transcendental, along with the theory of sensory integration, seeking to incorporate stimuli collected through the senses and neurological processes that integrate this information to generate adaptive responses in children to the demands of the environment. *‘In pediatrics, knowledge of psychomotor development will provide guidelines and directives for intervention.’**‘Theories of play and how children relate at a certain age, how they play at a certain age, why we decide to give them one toy and not another.’**‘The issue of sensory integration is also important to understand some behaviors of children.’*

From the knowledge in the socio-community area, the theory of popular education from Paulo Freire and its participatory methodologies emerge with strength, as well as community-based rehabilitation. This is mainly associated with primary health care devices from intersectoral work logics seeking to generate cooperation between community spaces such as neighborhood councils, schools, churches and clients and their families. 

The above is complemented by the human rights approach, understood as guarantees, liberties, and fundamental rights to human dignity, which guide and permeate occupational interventions seeking to generate conditions of equity for the subjects of care while also specializing in the rights of children, people with disabilities, migrants, and others related to occupational therapy work. 

Finally, the model of social networks appears, which seeks to establish links or connections through occupation between clients and their social support such as friends, family, school/work colleagues, but also with institutions to establish a network. *‘We use Paulo Freire’s strategies a lot, in other words, in this sense, popular education is an essential tool for us.’**‘I believe that the main thing is that occupational therapists themselves, as they are socio-community professionals, should work based on the network model, strengthening people’s networks and the health area from a functional rehabilitation point of view, with a human rights vision, I believe that this is fundamental.’**‘Intersectoriality is a very important issue because even though it can be a primary health care device, it has to do with the intersectoral, like, for example, the community-based rehabilitation strategy, which is a Primary Health Care strategy.’*

There is also a critique of training and specialization in community occupational therapy. *‘The whole world does groups and workshops, but they don’t know how they do it and where they get it or how they have been informed, and the same applies to community work. Now all the occupational therapists think they are community, but they don’t specialize in the area.’*

### 3.2. Skills

This is related to the practical skills and competencies that occupational therapists have. They are internalized and executed based on theoretical knowledge, but they also require creativity to adapt them to each user, as well as motor skills. They cover both the strategies and processes that are used in an orderly sequence of the therapeutic process, from assessment to elaborating clients’ occupational profiles, intervention design, and later follow-up (Figure 5). 

These competencies are linked to the use of occupation in rehabilitating people, which favors the occupational performance of users in their natural environments from occupational therapy programs. 

The evaluation process is holistic and is carried out through the application of specific evaluation guidelines and instruments, as well as clinical observation and interviews. It seeks to determine the functional independence of the person, evaluating both motor and sensory elements, as well as cognitive functions, activities of daily living, and the requirements of orthotics and/or technical aids. *‘Assessment guidelines should be managed to look at skills, manipulation as such; assessment guidelines for activities of daily living, postural control, joint range, muscle strength, muscle tone, turns, transfers, transitions, manual functionality, coordination, technical aids, and orthotics requirements and cognitive functions.’*

In pediatrics, elements of psychomotor evaluation, sensory integration, and play are also incorporated. *‘In pediatrics, you need knowledge of psychomotor development, motor development tests such as the Bayley Scale, the VMI, and sensory integration guidelines. It is required to have mastery of therapeutic play techniques, of how children of a certain age relate to each other, how to give them one type of toy or another, how to adapt it to their needs.’*

Assessment is carried out in the clinical context as well as in home visits, addressing family dynamics, physical and social environments in school and the workplace, and community diagnoses that aid with understanding and explaining the strengths, resources, and problems of a given community. *‘One must evaluate the dynamics of the parents since they are the ones who must replicate the therapies at home, as well as the knowledge about the pathology, the school, and the architectural barriers that are found in the spaces where the patient develops.’**‘The first stage mentioned in the diagnosis is to get to know each other, to get to know the community, who are part of, and this implies recognizing each other, where they come from, what their interests are, aspects that are basic to human communication and occupational therapy.’*

The intervention process consists of designing and implementing different activities and therapeutic strategies according to the needs detected in the evaluation considering the occupation, whose purpose is maximizing community inclusion of the subjects. 

This process seeks to strengthen or maintain motor, sensory, cognitive and communication skills, as well as spiritual and emotional elements, to improve the occupational performance of people in their self-care activities, productivity, and leisure time. 

Another important element is training in technical aids such as orthoses, wheelchairs, or other types of adaptations to the physical environment such as housing, the workplace, or school, if necessary to improve functional performance and autonomy. 

This is in addition to post discharge follow-up, which is gradually becoming more distant over time. *‘Treatment sessions where we work on training skills such as independence and indications of daily living, and training in technical aids, so it is often necessary to work previously on postural control, passive mobilization, manual functionality, muscle activity, and cognitive stimulation.’**‘Technical aids such as wheelchair training, transfer from bed to chair and from chair to shower, going up and down sidewalks. We take the patient out of the clinic to go shopping, take the elevator, go to the mall, university, school, work, and address architectural barriers that may arise.’*

Reference is made to the macrosocial space of intervention of the occupational therapist as well, around visions associated with the social structure and systems, to promote human relational and social capital. This type of intervention is related to social, cultural, and political components from a systemic point of view. *‘The intervention processes have to do with empowering the collective as a space for daily life, training, spaces of struggle, spaces of liberation, it is privileged, that is, being a citizen to the extent that you are part of a community, you are included, or you generate processes of inclusion in communities.’*

We found that there are other types of references associated with procedural competencies as well, such as community management, working with networks, coordination processes with other health professionals, other programs, and family members; and, to a lesser degree, research. *‘The entire process of managing both resources and coordination with other programs is important.’**‘Evidence-based occupational therapy is the class for neurorehabilitation processes as much from the clinical as from the community.’*

### 3.3. Attitude

These competencies are related to ethical components for decision making in therapeutic processes and are constituted by values, beliefs, and attitudes that govern the personal actions and social coexistence of the occupational therapist. 

These competencies vary depending on the community where intervention happens and on the characteristics of the occupational therapy professional who carries out the intervention, but they are centered on the therapeutic link, emphasizing personal characteristics such as empathy, applying the ethical principles of the profession, proactivity, and tolerance of frustration. 

The above adds to relational skills linked to teamwork, problem solving, respect for others, and communication skills, especially regarding the expectations of users, their families, and caregivers, as these must be mediated and accepted. *‘The most important thing in neurorehabilitation is teamwork. It is fundamental. And being able to listen to what the patient needs and wants, what motivations he has, being able to take these motivations and then move toward therapeutic goals.’*

In summary, Table 3 shows the most important competencies that a community occupational therapist in neurorehabilitation should have. 

## 4. Discussion

To perform community interventions in neurorehabilitation, an occupational therapist requires three essential competencies according to the perspective of the professionals interviewed. 

First, there are theoretical competencies related to biomedical and social sciences. An interesting finding relates to the necessity and importance of knowing the mechanisms of action of neurological pathologies and their treatments from the basics, understanding the interactions between people with a given health situation and their social environment, and generating interventions that favor personal and collective wellbeing in addition to achieving social inclusion. Therefore, incorporation of the occupational therapist as soon as possible in the rehabilitation process is of utmost importance, since together with this, the probability of readmission to the hospital system decreases [52]. 

The literature describes the models and/or strategies on which community interventions are based; however, it does not go into detail regarding the preparation or prior knowledge that professionals should have regarding the health situation of each client [23,24,25], which is fundamental for the participants in this study. 

It is stated that the interventions are based on evidence, but references are made that an important theoretical knowledge is the Bobath concept. However, the evidence collected in a systematic review with a meta-analysis proposed by Scrivener and collaborators [53] after stroke indicates that this therapy is less effective than, for example, task-specific training, and is not superior to other interventions, except for proprioceptive muscle facilitation. Likewise, in another literature review regarding interventions with children with cerebral palsy, the evidence indicates that it is ineffective in normalizing movement and in preventing contracture development, or the evidence is conflicting in relation to reduced spasticity or improved function [54]. Updating knowledge, as well as an effective search for information and a quality analysis of the available evidence, thus becomes an element of utmost importance for the practice in the field of neurorehabilitation. 

Now, in terms of theoretical knowledge of community work, the results coincide with those of Martínez and Guajardo, as it is understood that when using a community approach to guide therapeutic processes, there is a call to action to active participation of people and institutions, improving the symptomatology and health situation of people using their resources and those of the territory [16,17].

However, some participants mentioned that in practice, most occupational therapists use participatory strategies to work from a community approach, but it is suggested that there is little theoretical knowledge to support it. Therefore, the professionals in the area must specialize in the community approach and begin to systematize experiences, including components widely recognized as key to biomedical and social sciences. 

Second, procedural competencies are needed, which are used in the execution of evaluations, interventions, and follow-up of therapeutic processes, in addition to management and research skills. Studies aimed at recording and evaluating experiences in community work have focused mainly on describing this type of competency [23,24,25,26].

The results of this investigation highlight the biomedical sciences procedures with more clarity and specificity than those of the social or community sciences. The work of the occupational therapists interviewed focuses on interventions at the personal and micro-social level, since they are centered on working with users and their close social environment such as family, school, and work colleagues, as well as in the physical environment of home, school, and the workplace. 

Although the final objective of community intervention is present in the discourse to achieve a structural change regarding inclusion in all the spheres in which people work, the results do not show concrete strategies for practices at the macrosocial level. The latter, pointed out by Ulloa and Pino [24], could be detrimental to the rehabilitation processes of people whose health status depends mainly on structural changes in society. 

Finally, the attitudinal competencies described in this study, associated with ethical elements, teamwork, respect, communication, active listening, among others, are similar to those reflected in the studies of Öst Nilsson et al. [29], Kendall et al. [31] and Gharib et al. [30].

The identification of these three types of competencies is consistent with what has been proposed in other research [6,7,55] on health competencies, which consists of the relationship between knowledge or theoretical and technical competencies, know-how or procedural or methodological competencies, and know-how to be or attitudinal or personal competencies. 

Other practical applications that this research can provide are the contribution to occupational therapists’ training. Undergraduate programs can also consider these competencies to nurture their curricula, for example, the community occupational therapy subject.

It is important to consider the results obtained in this investigation. They come from the practical experience of occupational therapists working in neurorehabilitation and community intervention and can be used as elements in the training of future professionals or as a guide for professionals recently related to the area. However, due to the range of work experience years among the occupational therapists interviewed, reaching from 4 to 22 years, their intrinsic opinion is influenced by the experience and expertise achieved through their professional career, which generates differences in participants’ opinions. It would be interesting to address this phenomenon in future research, since one of the limitations of this study was that the data analysis did not consider this variability.

In addition, there is another limitation in relation to the methodology used, since from its phenomenological and qualitative positioning, it contemplates and visualizes a local reality of the phenomenon, from its actors through the sample focused only on Chile, such that the results cannot necessarily be extrapolated to all Ibero-American or international realities. Furthermore, the scope of this study is descriptive and does not allow us to generate conjectures at an explanatory or correlational level. 

Various intriguing new questions arise from this research. Does this approach coincide with others at an international level? Is it agreed upon by all the participants? Could the users’ perception of occupational therapists’ work provide us with other information regarding their competencies? New avenues of research are thus opened by other types of qualitative methodologies such as focus groups, or consensus methods with experts and stakeholders such as the Delphi method, to generate consensus on competencies. This study can be used as a basis for developing quantitative studies as well, such as applying surveys that can verify the acceptance or non-acceptance of the elements proposed by the occupational therapist community. It is also suggested to incorporate the perceptions of these services’ users, which can be contrasted with this study. 

## 5. Conclusions

The community approach to neurorehabilitation requires the presence of three types of professional competencies in occupational therapists: theoretical, procedural, and attitudinal competencies, which are put into practice throughout the intervention process. These competencies are fundamental to carry out effective interventions with users and their communities, thereby favoring optimal occupational performance and social participation. 

These neurorehabilitation processes should be based on the problems identified by the client, to improve and/or maintain their physical, psychological, social, and emotional health, with the goal of social inclusion within family, school, or work, as well as with the community and society itself.

Working with a community approach means conceiving the therapeutic process integrally. It is not limited to social aspects, but it also includes the theoretical bases for the physical and mental health situation of individuals and communities. Each intervention is required to have strong theoretical support from the biomedical and social sciences. 

To achieve an effective process, it is necessary to intervene from a micro- and macro-social level as well. At the microsocial level, interventions must focus on clients and their immediate environments, and at the macrosocial level, they must facilitate structural and institutional conditions at a higher level of public policy, whether local, regional, or national, to guarantee universal accessibility.

## Figures and Tables

**Figure 1 ijerph-19-06096-f001:**
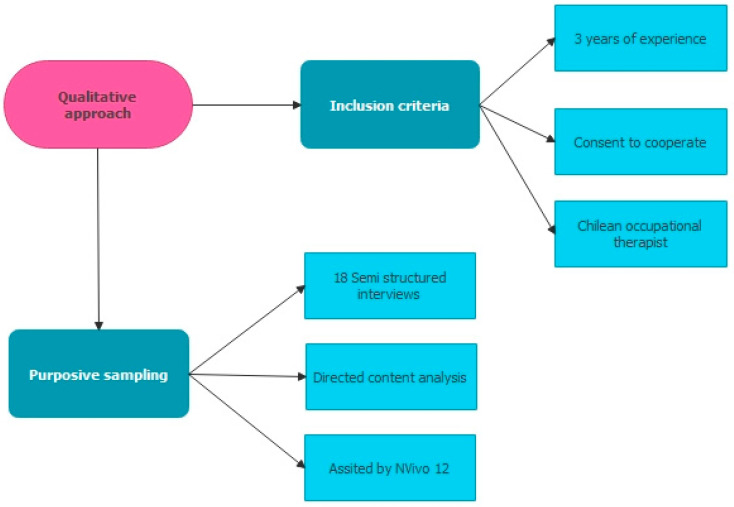
Design of the investigation.

**Figure 2 ijerph-19-06096-f002:**
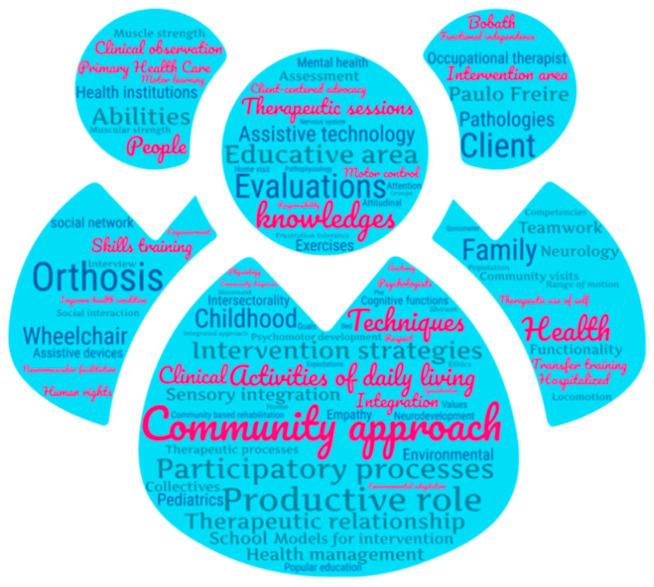
Word cloud of interviewee discourse. Random visualization of the 100 most frequent words and phrases in participants’ discourse. The size of each word is proportional to the number of times it was counted in the text.

**Figure 3 ijerph-19-06096-f003:**
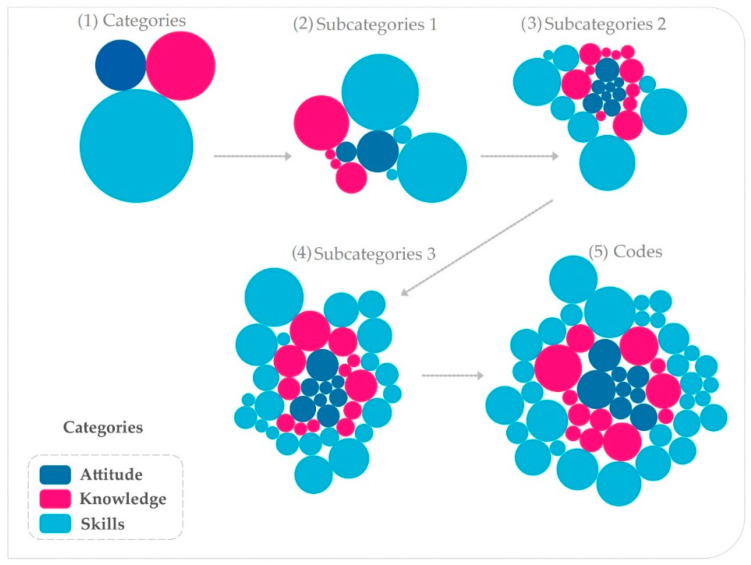
Operationalization of qualitative variables. The analysis considered five levels of analysis, from 3 categories to 51 codes. The higher levels were contained in the lower levels. Analysis complexity increases and is broken down into (1) 3 categories: (2) 10 subcategories level one, (3) 27 subcategories level two, (4) 42 subcategories level three, and (5) 51 codes.

**Figure 4 ijerph-19-06096-f004:**
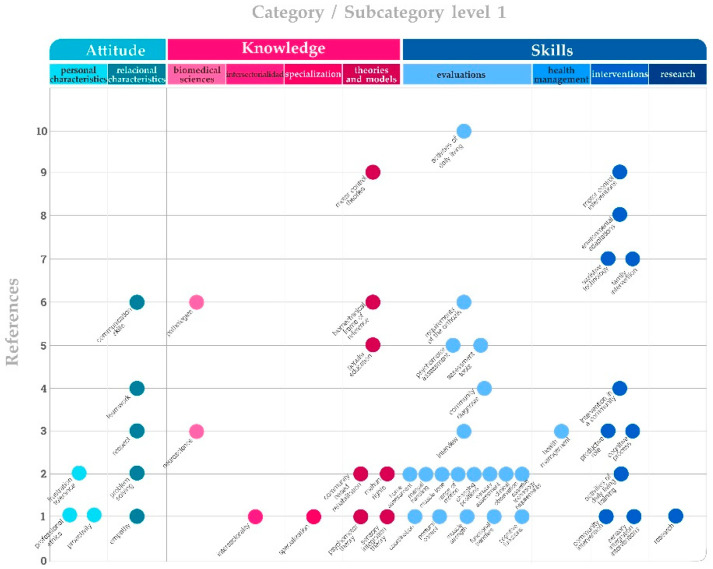
Diagram of code references. Each circle represents a code associated with ten subcategory levels, which are linked to the three major categories: attitude, knowledge and skill.

**Figure 5 ijerph-19-06096-f005:**
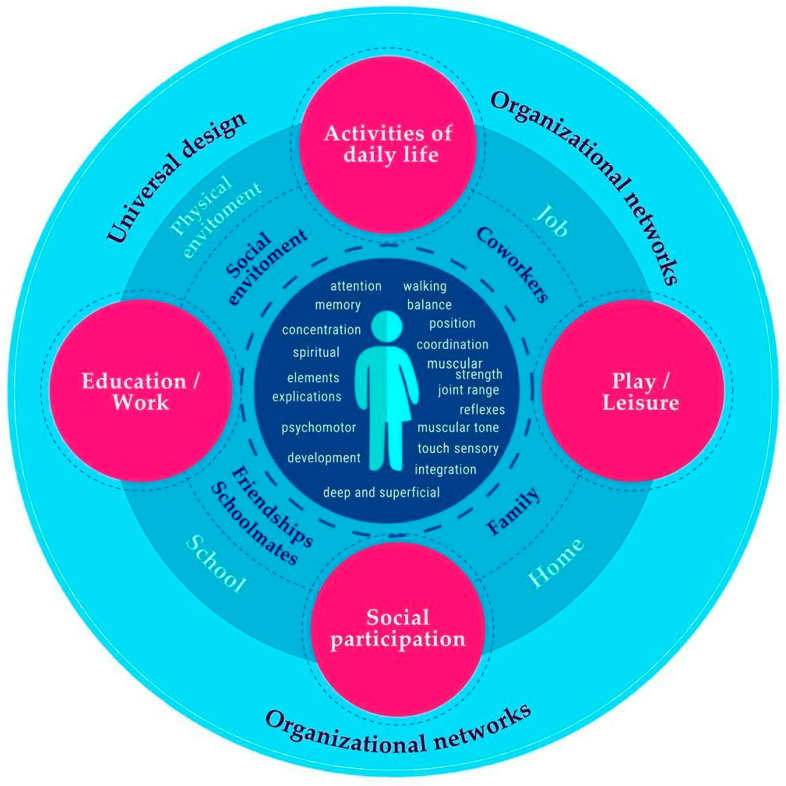
A community approach to neurorehabilitation. This figure represents the levels of approach and practical competencies of community occupational therapy intervention in neurorehabilitation, focusing on daily life occupations, study/work, play and leisure time, as well as social participation. It addresses elements at the personal level, represented by green, and in the biological, psychological, and spiritual dimensions and at the micro-level, represented by the color light blue. At the macro level, the color apricot highlights aspects of accessibility, community, and organizational networks.

**Table 1 ijerph-19-06096-t001:** Description of study participants.

Participant	Sex	Age	Care Population Type	Care	Device Type
1	Female	34	Juvenile/Adult/Senior	Primary	Public
2	Female	31	Juvenile	Secondary/Tertiary	Public
3	Female	35	Juvenile/Adult/Senior	Primary	Public
4	Male	26	Juvenile/Adult/Senior	Primary	Public
5	Male	32	Adult/Senior	Primary	Public
6	Female	40	Adult/Senior	Secondary	Private
7	Female	55	Juvenile/Adult/Senior	Secondary	Private
8	Female	36	Juvenile/Adult/Senior	Primary	Public
9	Female	46	Juvenile/Adult/Senior	Primary	Public
10	Male	26	Juvenile/Adult/Senior	Primary	Public
11	Female	35	Adult/Senior	Secondary	Private
12	Female	31	Adult/Senior	Secondary	Private
13	Male	42	Juvenile/Adult/Senior	Primary	Public
14	Female	31	Juvenile/Adult/Senior	Primary	Public
15	Female	44	Juvenile	Secondary/Tertiary	Private
16	Female	42	Juvenile	Secondary/Tertiary	Private
17	Female	33	Adult	Secondary	Private
18	Female	41	Adult	Secondary	Private

**Table 2 ijerph-19-06096-t002:** Semi-structured interview script.

Dimension	Questions
Knowledge	What theories or models do you habitually use? What knowledge should an occupational therapist have when working in neurorehabilitation? What theoretical background do you associate with community work?
Skills	What procedures do you habitually do? What type of evaluations do you use in your practice? How do you do them? Can you describe the interventions you typically do? How do you link your intervention with community space? What community actions do occupational therapists do in your workspace?
Attitudes	What values predominate in OT practice in your area? What attitudes do you believe to be essential for your neurorehabilitation work

**Table 3 ijerph-19-06096-t003:** Summary of the competencies of a community neurorehabilitation occupational therapist.

Type	Main Competencies
Theoretical knowledge	Neuroscience Pathologies Biomechanical Frame of Reference Sensory Integration Theory Psychomotor Theory Community-Based Rehabilitation Human Rights Popular Education Social Network Theory
Procedural content	Environmental Assessment Sensorimotor Functions Psychomotor Assessment Sensory Assessment Interview Clinical Observation Psychomotor Assessment Sensory Assessment Interview Clinical Observation Assessment Tools
Attitudinal content	Empathy Communication Skills Problem Resolution Respect Teamwork Frustration Tolerance Professional Ethics Proactivity

## Data Availability

The data presented in this study are available on request from the corresponding author.

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
