# Peer review of "What Competencies Does a Community Occupational Therapist Need in Neurorehabilitation? Qualitative Perspectives"

_ijerph, 2022, doi:10.3390/ijerph19106096_

Round 1

Reviewer 1 Report

The main results of this article the article is associated to the competencies of Knowledge regarding 21 biomedical and community theoretical elements. Although the scope of the study is limiter; I still think it may be worth to be published. However, I do not think holistic approaches are good enough to conduct research studies in general.

This observation may not suit this study because it is influenced by the my research technological background in this area which is not the area of the study presented in this paper.

I think the paper is not difficult to read and it is well-structured, but conclusions are too short. I encourage authors to extend them (conclusions are written in a third of a page for 16 pages paper). I would also suggest fixing the format of Figure 2and 3 (they seem too large).

Finally, about writing, I think that authors should perform a full review of the article employing a native speaker. In addition, I would suggest changing “neurorehabilitation” to “Neurorehabilitation” in the title of the article.

Author Response

Dear Reviewer

We appreciate your time in reviewing our manuscript and providing your valuable comments. We hereby respond to the proposals of the reviewer for the article titled ‘What competencies does a Community Occupational Therapist Need in Neurorehabilitation? Qualitative Perspectives from Practice’, which are detailed below through the following table:

Reviewers' comments considered

Modifications and adjustments by authors

Reviewer 1

 1.- In the title, change “neurorehabilitation” to “Neurorehabilitation.”

 This suggestion is accepted and the text was modified.

2. It is suggested to extend the conclusions in relation to the number of pages of the paper.

The suggestion is accepted, the conclusions were extended in relation to the results and discussion of our article. The depth of these issues was increased.

3.- It is suggested to modify the size of figures 2 and 3, since they are too large.

 The size of Figures 2 and 3 was modified to a smaller size in proportion to the document.

4.- It is suggested that the language of the whole text be revised by a native speaker.

The request was accepted, the entire text was revised by a professional English-Spanish translator and a native speaker (Annexed 1 in the attachment document).

We look forward to hearing from you and are willing to respond to any further questions and comments you may have.

Sincerely,

Daniela Avello-Sáez

Fabiola Helbig-Soto

Nayadet Lucero-González

María del Mar Fernández-Martínez

Reviewer 2 Report

Thank you for allowing me to read this paper which initially deals with an interesting and relevant topic in OT. The suggestions given are intended to improve your work.

  • Title: I think there should be some reference in the title to the fact that this is a qualitative research study.
  • Check the numbering of headings and subtitles. There are some mistakes.
  • Introduction: Although the introduction is interesting, it is a bit disorganised. It might be interesting for them to define competences in more detail. Perhaps the authors could organise their ideas a little more: There is a part where they talk about the importance of the community approach, another part about what OT is, ... I extend this recommendation to all work. Try to reorganise your ideas better. I think that the use of subtitles in the introduction and discussion sections is a good idea.
  • Paulo Freire's theory of popular education and its participatory methodologies: I think this should be explained better, it is not something readers should be familiar with, bearing in mind that this article will mainly be of interest to occupational therapists (now it is only in results section).
  • Participants: Sociodemographic characteristics of the sample should be explained in much more detail. How was the sampling performed?
  • Ethical concerns: It is extremely important that you provide a code of ethics committee approval for the conduct of the study (this is grounds for refusal).
  • Results:
    • “In general, it can be indicated that these categories are founded on Theoretical Competencies of health conditions, as well as the community approach; Procedural Competencies regarding the evaluation and intervention on the personal level, in interaction with their physical environment, as well as on a micro and macro social level; and Procedural Competencies, related to personal and relational characteristics of the occupational therapist.” Please, add references or explain this better, it is not clear why this is the basis for this - are there no attitudinal competences? Later in the text, the authors talk about them.
    • “Competencies were grouped into three main categories based on bibliographical review. Please, add references.
    • Figure 2: If the authors could insert the names of the variables/codes in this graph, it would be great to make the article easier to read.
    • I would like to throw a thought to the authors. In the results, in the category of knowledge, you talk about evidence-based practice. However, they then give this example: “It is important for neurorehabilitation to know about motor control, activity-oriented motor 269 learning, neuronal plasticity, neuromuscular facilitation, the Bobath concept, which is the one that 270 is most used”.Considering that the current evidence does not support the Bobath approach, what fit, or explanation would this have in your work?
      • https://onlinelibrary.wiley.com/doi/pdf/10.1111/j.1469-8749.2007.00723.x
      • https://www.sciencedirect.com/science/article/pii/S183695532030103X?via%3Dihub
    • “Competencies were grouped into three main categories based on bibliographical review. Please, add references.
    • The part on attitudinal competences is weak - can you add something more?
  • Discussion could be improved. I believe that theoretical and practical applications should be more clearly reflected.
  • Conclusions: Please summarise in this section ONLY the main findings of your work.
  • Citations and References: I believe there are statements throughout the paper that require bibliographical citations. Please review the whole text.

Author Response

Dear Reviewer: 

We appreciate your time in reviewing our manuscript and providing your valuable comments. We hereby respond to the proposals of the reviewer for the article titled ‘What competencies does a Community Occupational Therapist Need in Neurorehabilitation? Qualitative Perspectives from Practice’, which are detailed below through the following table:

Reviewers' comments considered

Modifications and adjustments by authors

1.- Title: I think there should be some reference in the title to the fact that this is a qualitative research study.

The new title is: What Competencies does a Community Occupational Therapist Need in Neurorehabilitation? Qualitative Perspectives from Practice.

2.- Check the numbering of headings and subtitles. There are some mistakes.

We checked the numbering of headings and corrected them.

·         3.- Introduction: Although the introduction is interesting, it is a bit disorganized.

·         It might be interesting for them to define competencies in more detail.

Try to reorganize your ideas better. I think that the use of subtitles in the introduction and discussion sections is a good idea.

We included the subtitles in the introduction.

We included the definition of competencies and occupational therapy and were more descriptive.

4.- Paulo Freire's theory of popular education and its participatory methodologies: I think this should be explained better, it is not something readers should be familiar with, bearing in mind that this article will mainly be of interest to occupational therapists (now it is only in results section).

We added an explanation and bibliography about popular education from Paulo Freire.

5.- Participants: Sociodemographic characteristics of the sample should be explained in much more detail. How was the sampling performed?

The Materials and Methods section were extended, and the topics such as sociodemographic and sampling were added with more description and with more details.

6.- Ethical concerns: It is extremely important that you provide a code of ethics committee approval for the conduct of the study (this is grounds for refusal).

The ethical concerns were described with more specifications; however, because this type of research does not require an ethical committee, as the Code of Good Research Practice of the University of Almería (Annexed 2 in the attachment document).

·         7.- Results:

“In general, it can be indicated that these categories are founded on Theoretical Competencies of health conditions, as well as the community approach; Procedural Competencies regarding the evaluation and intervention on the personal level, in interaction with their physical environment, as well as on a micro and macro social level; and Procedural Competencies, related to personal and relational characteristics of the occupational therapist.” Please, add references or explain this better, it is not clear why this is the basis for this - are there no attitudinal competences? Later in the text, the authors talk about them.

The second time Procedural Competencies was cited, it had a mistake because it must be attitudinal competence.

It was changed in the document.

·         8.- “Competencies were grouped into three main categories based on bibliographical review. Please, add references.

These references were added.

·         9.- Figure 2: If the authors could insert the names of the variables/codes in this graph, it would be great to make the article easier to read.

The names of variables were added in Figure 2

·         10.- I would like to throw a thought to the authors. In the results, in the category of knowledge, you talk about evidence-based practice. However, they then give this example: “It is important for neurorehabilitation to know about motor control, activity-oriented motor 269 learning, neuronal plasticity, neuromuscular facilitation, the Bobath concept, which is the one that 270 is most used”.Considering that the current evidence does not support the Bobath approach, what fit, or explanation would this have in your work?

The data collection was contrasted with the evidence provided in the discussion item. Actually, this information was incorporated into the references too.

·         11.- The part on attitudinal competences is weak - can you add something more?

The attitudinal competencies were minor in the data analysis, and for that, the representation in the manuscript is shorter than the other two.

·         12.- Discussion could be improved. I believe that theoretical and practical applications should be more clearly reflected.

The discussion was improved, and we added a paragraph about practical applications.

·         13.- Conclusions: Please summarise in this section ONLY the main findings of your work.

An attempt was made to prioritize the most relevant elements. However, they tried to conjure with what was suggested by reviewer 1, who requested to expand this section.

·         14.- Citations and References: I believe there are statements throughout the paper that require bibliographical citations. Please review the whole text.

The whole text was reviewed, and we added the citations where we believe they might be missing.

We look forward to hearing from you and are willing to respond to any further questions and comments you may have.

Sincerely,

Daniela Avello-Sáez

Fabiola Helbig-Soto

Nayadet Lucero-González

María del Mar Fernández-Martínez

Reviewer 3 Report

I would like to thank you for submitting and give me the opportunity to review the manuscript entitled: " What Competencies does a Community Occupational Therapist Need in neurorehabilitation? Perspectives from Practice". The research topic undertaken by the authors is interesting and reveals competencies of high importance for the interviewed therapists that have not been collected in previous studies so far, which can be used as a basis for the development of quantitative studies, absolutely necessary in this discipline. The study is well performed, the results are compelling and adequately presented. This study should be of interested for the journal readers. Nevertheless, some questions and concerns need to be answered and corrected before the formal acceptance of the manuscript. In this sense, I only have a few minor comments. I hope my comments will help to improve the quality of the manuscript in some way.

Since this is a mainly qualitative opinion study, it is necessary to detail whether the same researchers conduct the interviews and analyze the data. It is important not to commit bias in the interpretation of data. On the other hand, how do you ensure the interviewer's objectivity during the interviews?

The approval of the ethics committee of the institution where the study is carried out is not stated in the text. If this study is exempt from this requirement, it is necessary to add it in the text, and if not, it is necessary to include the registration number of approval by the ethics committee. This is a very important consideration.

Since this is a mainly qualitative opinion study, it is necessary to detail whether the same researchers conduct the interviews and analyze the data. It is important not to commit bias in the interpretation of data. On the other hand, how do you ensure the interviewer's objectivity during the interviews?

Regarding the results section, in the information in the figure caption within the category of knowledge, in the subcategory of theory and models, would it not be 7 codes instead of 5 codes due to the fact that there are 7 red circles? If this is the case, I recommend modifying the text. The same observation concerning the skills category, where the text states that “subcategory of Evaluations with 10 codes ranging from 1 to 9 references” and there are 19 blue circles raging from 1 to 19 references. Please, check it.

Following the same figure, I recommend following the same guidelines in all subcategories. You write the range of references in all of them except in personal characteristics. I recommend checking and specifying them in the figure caption.

In table 1 and for a better and quicker visualisation, I suggest to put dividing lines between the different competences.

Regarding the wide range in years of work experience (4 to 22 years), do you think that this large variability in time worked may affect the responses to the questionnaire? In other words, is it possible that workers' opinions are more influenced by the time worked than by the worker's own intrinsic opinion, biasing the responses?

Since this is a mainly qualitative opinion study, it is necessary to detail whether the same researchers conduct the interviews and analyze the responses. It is important not to commit bias in the interpretation of data. On the other hand, how do you ensure the interviewer's objectivity during the interviews?

Although it is mentioned in the limitations of the study, I consider that both the perception of the users and not only of the workers would have provided very relevant information to the study. Likewise, quantitative data on the responses to the questionnaire would also have enriched the work. In any case, as you rightly say, this work can serve as a basis for the development of future quantitative work in the discipline.

Finally, some unimportant errors, the reference [39] on line 180 appears to be in a different font (format) from the rest of the text. Check it out.

Author Response

Dear Reviewer: 

We appreciate your time in reviewing our manuscript and providing your valuable comments. We hereby respond to the proposals of the reviewer for the article titled ‘What competencies does a Community Occupational Therapist Need in Neurorehabilitation? Qualitative Perspectives from Practice’, which are detailed below through the following table:

Reviewers' comments considered

Modifications and adjustments by authors

Reviewer 3

1.- Since this is a mainly qualitative opinion study, it is necessary to detail whether the same researchers conduct the interviews and analyze the data. It is important not to commit bias in the interpretation of data. On the other hand, how do you ensure the interviewer's objectivity during the interviews?

The suggestion is accepted and information is added to the methodology.

For information recollection, a semi structured in-depth interview guideline [40] was validated by a group of experts in the field.

One of the researchers was responsible for conducting the interviews with the participants, and the analysis by two researchers separately. Finally, the results were discussed by the entire research team.

The interviews were carried out at the interviewee's workplace with an approximate duration of 90 minutes each, and a field note was prepared for each participant. To ensure objectivity in the interpretation of the data, the results were discussed with the research team.

2.- The approval of the ethics committee of the institution where the study is carried out is not stated in the text. If this study is exempt from this requirement, it is necessary to add it in the text, and if not, it is necessary to include the registration number of approval by the ethics committee. This is a very important consideration.

The suggestion was accepted, and information was added.

Likewise, it is important to note that this study, due to its qualitative methodology and the instrument used, was exempt from the requirement to be approved by an ethics committee.

3.- Regarding the results section, in the information in the figure caption within the category of knowledge, in the subcategory of theory and models, would it not be 7 codes instead of 5 codes due to the fact that there are 7 red circles? If this is the case, I recommend modifying the text. The same observation concerning the skills category, where the text states that “subcategory of Evaluations with 10 codes ranging from 1 to 9 references” and there are 19 blue circles raging from 1 to 19 references

You write the range of references in all of them except in personal characteristics. I recommend checking and specifying them in the figure caption.

The description of the figure is modified.

Figure 3. Diagram of code references. Each circle represents a code. Knowledge groups subcategories of Biomedical Sciences with 2 codes ranging from 3 to 6 references, Theories and Models with 57 codes ranging from 1 to 9 references, Specialization and Intersectoriality with 1 reference. Skills groups the subcategories of Evaluations with 1019 codes ranging from 1 to 9 references, Interventions with 10 codes ranging from 1 to 9 references, and Health Management and Research with 1 code, respectively. The last category, Attitude has Relational Characteristics with 5 codes, ranging from 1 to 6 references, and Personal Characteristics with 3 codes, ranging from 1 to 3 references.

4.- In table 1 and for a better and quicker visualisation, I suggest to put dividing lines between the different competences.

The suggestion was accepted and the table was modified.

5.- Regarding the wide range in years of work experience (4 to 22 years), do you think that this large variability in time worked may affect the responses to the questionnaire? In other words, is it possible that workers' opinions are more influenced by the time worked than by the worker's own intrinsic opinion, biasing the responses?

This suggestion was accepted and information was added.

It is important to consider the results obtained in this investigation, as they come from the practical experience of occupational therapists working in neurorehabilitation and community intervention, as they may be used as elements in the training of future professionals or as a guide for professionals recently related to the area. However, due to the variability of years of work performance of the occupational therapists interviewed, ranging from 4 to 22 years, their intrinsic opinion is influenced by the experience and expertise achieved through their professional career, which entails a difference in the opinions of the participants. It would be interesting to address this phenomenon in a future research, since one of the limitations of this study was that the data analysis did not consider this variability

6. Although it is mentioned in the limitations of the study, I consider that both the perception of the users and not only of the workers would have provided very relevant information to the study. Likewise, quantitative data on the responses to the questionnaire would also have enriched the work. In any case, as you rightly say, this work can serve as a basis for the development of future quantitative work in the discipline

This suggestion was accepted and information was added.

New questions arise from this research, such as: does this approach coincide with others at an international level? Is it agreed upon by all participants? And could the users' perception of the work of occupational therapists provide us with other information regarding their competencies?

We look forward to hearing from you and are willing to respond to any further questions and comments you may have.

Sincerely,

Daniela Avello-Sáez

Fabiola Helbig-Soto

Nayadet Lucero-González

María del Mar Fernández-Martínez

Round 2

Reviewer 1 Report

Author improvements have a strong impact in the quality of the new version of the paper.

Although I still think that the impact in the IT community is really low; it may be published in the current form.

Author Response

Thank you for providing your comments. 
We have extensively improved the methods and better explained the research method. We hope that you find our revised version a greater contribution to the discipline. 

Reviewer 2 Report

Thank you for allowing me to read this paper which initially deals with an interesting and relevant topic in OT. The suggestions given are intended to improve your work.

The main problem remains the organisation of the text, which makes it complex to follow.

  • Introduction: The first part doesn’t have a subheading. There are still mixed ideas. As it is now presented, a first point could be about neurodological diseases, a second one about the characteristics of the health system and the prevalence in Chile, a third one about the importance of OT competences and another one about community experiences (this par maybe should be summarized). I think that would make it clearer.
  • The introduction should end with a clear objective.
  • Materials and methods structure: please organise this section with the sub-sections it should have study design, ethical considerations, participants, instruments, procedure, statistical analysis...
  • Participants: Sociodemographic characteristics of the sample should be explained in much more detail (year, sex, years of practise, expertise… all you can add). Perhaps a table should help. How was the sample recruited? Which where the eligibility criteria?
  • Ethical concerns: Code of Good Practice of the International PhD School at the University of Almería, Spain. Could you provide this document, please?
  • Results: The first part doesn’t have a subheading.
  • Discussion could be improved. It cannot begin as it begins. Please provide 1) the main findings of your research, 2) analysis of your results compared to other research, 3) practical implications, 4) limitations and 5) future lines of action.
  • Conclusions: Please summarise in this section ONLY the main findings of your work.
  • References: follow the format recommended by the journal.

Author Response

Thank you for carefully reading our manuscript. We have extensively edited the entire thing, to make it easier to read. We answer your comments below in bold text. 

The main problem remains the organisation of the text, which makes it complex to follow.

We reorganized a large portion of the text, summarized and rewrote major parts of the paper to make it easier to read.  

  • Introduction: The first part doesn’t have a subheading. There are still mixed ideas. As it is now presented, a first point could be about neurodological diseases, a second one about the characteristics of the health system and the prevalence in Chile, a third one about the importance of OT competences and another one about community experiences (this par maybe should be summarized). I think that would make it clearer.

We put subheadings in the introduction, per the reviewer's request. We also summarized and cut the introduction down, to make it easier to read. Thank you for pointing this out. 

  • The introduction should end with a clear objective.

The research objective was added to the end of the introduction. 

  • Materials and methods structure: please organise this section with the sub-sections it should have study design, ethical considerations, participants, instruments, procedure, statistical analysis...

We completely rewrote the methods section and added each of the sub-sections suggested by the reviewer. Each part should now be quite clear.

  • Participants: Sociodemographic characteristics of the sample should be explained in much more detail (year, sex, years of practise, expertise… all you can add). Perhaps a table should help. How was the sample recruited? Which where the eligibility criteria?

We added extensive information, including a table about the participants, including all the information suggested by the reviewer. We also added information about the recruitment process and eligibility criteria, per the reviewer's comments. 

  • Ethical concerns: Code of Good Practice of the International PhD School at the University of Almería, Spain. Could you provide this document, please?

We have provided this document in the supplementary materials. 

  • Results: The first part doesn’t have a subheading.

We added a subheading and re-organized the results according to the reviewer's comments. 

  • Discussion could be improved. It cannot begin as it begins. Please provide 1) the main findings of your research, 2) analysis of your results compared to other research, 3) practical implications, 4) limitations and 5) future lines of action.

We have rewritten the discussion per the reviewer's observations. It now includes the main findings, compares our results with other studies, gives practical implications, limitations and future lines of action. We hope that the reviewer finds our edited version a much better organized paper. 

  • Conclusions: Please summarise in this section ONLY the main findings of your work.

We have summarized the conclusions, and cut them down so that ONLY the main findings of our work appear in this section. 

  • References: follow the format recommended by the journal.

We have revised the format, and made changes to follow the journal's guidelines on format. 

Thank you for your comments, we hope that our revised version is of the reviewer's liking. We have extensively edited the manuscript, considering each of the reviewer's observations, and we feel the revised version is much better.